# Total escape of SARS-CoV-2 from dual monoclonal antibody therapy in an immunocompromised patient

Lena Jaki [1,9], Sebastian Weigang [1,9], Lisa Kern [1,9], Stefanie Kramme[2], Antoni G. Wrobel [3], Andrea B. Grawitz[4], Philipp Nawrath[3], Stephen R. Martin[3], Theo Dähne [1], Julius Beer[1], Miriam Disch[1], Philipp Kolb [1], Lisa Gutbrod[1], Sandra Reuter [2], Klaus Warnatz [5,6], Martin Schwemmle [1], Steven J. Gamblin [3], Elke Neumann-Haefelin[7], Daniel Schnepf [1], Thomas Welte[7,8], Georg Kochs [1], Daniela Huzly[1], Marcus Panning [1] ✉ & Jonas Fuchs [1] ✉

Monoclonal antibodies (mAbs) directed against the spike of severe acute respiratory syndrome coronavirus 2 (SARS-CoV-2) are effective therapeutic options to combat infections in high-risk patients. Here, we report the adaptation of SARS-CoV-2 to the mAb cocktail REGN-COV in a kidney transplant patient with hypogammaglobulinemia. Following mAb treatment, the patient did not clear the infection. During viral persistence, SARS-CoV-2 acquired three novel spike mutations. Neutralization and mouse protection analyses demonstrate a complete viral escape from REGN-COV at the expense of ACE-2 binding. Final clearance of the virus occurred upon reduction of the immunosuppressive regimen and total IgG substitution. Serology suggests that the development of highly neutralizing IgM rather than IgG substitution aids clearance. Our findings emphasise that selection pressure by mAbs on SARS-CoV-2 can lead to development of escape variants in immunocompromised patients. Thus, modification of immunosuppressive therapy, if possible, might be preferable to control and clearance of the viral infection.

It has become apparent that SARS-CoV-2, the causative agent of coronavirus disease 19 (COVID-19), can lead to persistent infections in immunocompromised individuals[1-5]. Several studies have reported viral intra-host evolution in such patients resulting in mutations that could be linked to immune escape[6-8]. Although vaccinations are beneficial, patients with weakened immune systems show an impaired immune response and are at high risk for severe COVID-19[9-11]. Therapeutic neutralizing mAb targeting the viral spike protein (S) are effective treatment options for such high-risk patients[12-14]. These includes the use of single neutralizing mAbs like Sotrovimab and

[1]Institute of Virology, Freiburg University Medical Center, Faculty of Medicine, University of Freiburg, Freiburg, Germany. [2]Institute for Infection Prevention and Hospital Epidemiology, Freiburg University Medical Center, University of Freiburg, Freiburg, Germany. [3]The Structural Biology of Disease Processes Laboratory, The Francis Crick Institute, London, UK. [4]Institute for Clinical Chemistry and Laboratory Medicine, Freiburg University Medical Center, Faculty of Medicine, University of Freiburg, Freiburg, Germany. [5]Department of Rheumatology and Clinical Immunology, Freiburg University Medical Center, Faculty of Medicine, University of Freiburg, Freiburg, Germany. [6]Center for Chronic Immunodeficiency (CCI), Freiburg University Medical Center, Faculty of Medicine, University of Freiburg, Freiburg, Germany. [7]Renal Division, Department of Medicine, Freiburg University Medical Center, Faculty of Medicine, University of Freiburg, Freiburg, Germany. [8]Friedrich Miescher Institute for Biomedical Research, Basel, Switzerland. [9]These authors contributed equally: Lena Jaki, Sebastian Weigang, Lisa Kern. ✉e-mail: marcus.panning@uniklinik-freiburg.de; jonas.fuchs@uniklinik-freiburg.de

Bamlanivimab or antibody cocktails such as REGN-COV, which consists of equal amounts of Casirivimab and Imdevimab. In cell culture systems, selection pressure by single mAbs can cause rapid evolution of SARS-CoV-2[15–18] suggesting that this might also occur in treated patients. Indeed, mutational escape from single therapeutic mAbs has been reported in patients after treatment with Bamlanivimab or Sotrovimab[19,20]. Therefore, therapeutic mAb cocktails like REGN-COV are used to avoid the emergence of SARS-CoV-2 escape variants[21,22]. Although quite successful in non-immunocompromised COVID-19 patients, it is not known whether treatment with mAb cocktails successfully suppresses mutational escape in immunocompromised individuals with a prolonged SARS-CoV-2 infection.

Here, we describe two kidney transplant patients infected simultaneously with the same SARS-CoV-2 Delta variant in the course of a nosocomial outbreak. Both were treated with REGN-COV at the same time after diagnosis. While one rapidly cleared the infection within days, the other patient had a persistent infection and virus was isolated from multiple oropharyngeal swabs. During prolonged replication three novel mutations emerged in the viral S gene, with late virus isolates showing a near complete escape from the REGN-COV treatment. This suggests a strong selection pressure on the S gene caused by the therapeutic mAbs in the absence of a functional adaptive immune system. Viral clearance only occurred after modification of the immunosuppressive therapy, thereby preventing the possible forward transmission of a virus variant resistant to one of the major therapeutic intervention strategies to treat COVID-19.

## Results

### Infection of two kidney transplant patients within a nosocomial SARS-CoV-2 outbreak at the University Medical Centre Freiburg, Germany

In late 2021, we observed a nosocomial outbreak at the University Medical Centre Freiburg that included ten cases based on likely transmission events due to direct contacts or spatiotemporal proximity (Fig. 1a). All patients tested negative for SARS-CoV-2 in the mandatory qPCR-based patient screening preceding their hospital admission. The suspected index case was a non-vaccinated inpatient ('case A') likely infected by an unvaccinated visitor who tested positive 2 days after the visit. Subsequently, we identified nine possible contact cases (three unvaccinated) in three different hospital wards who developed laboratory confirmed SARS-CoV-2 infections over a time period of two weeks. Four of them had severe COVID-19 (one died) whereas the remaining patients were asymptomatic or had mild respiratory symptoms (Supplementary Table 1). We performed whole-genome sequencing of SARS-CoV-2 from respiratory specimens of these patients and constructed a phylogenetic tree with sequences derived from other patients at the Medical Centre in late 2021 (Fig. 1b). Interestingly, all but one ('case B') of the nosocomial viral sequences clustered closely in the analysis indicating two separate transmission chains. The main outbreak was caused by the SARS-CoV-2 AY.43 lineage, a European Delta lineage with significant spread in 2021.

Two patients of this outbreak were kidney transplant recipients receiving immunosuppression (case H and I, here after referred to as 'patient 1' and 'patient 2'). Patient 1 (female, 70 years old, three-times vaccinated with mRNA vaccine) was hospitalized for elective hip replacement surgery and patient 2 (female, 61 years old, four-times vaccinated with mRNA vaccine) (Supplementary Table 1) for urinary tract infection (Fig. 2a, b, Supplementary Fig. 1b). Patient 1 had received a kidney allograft with induction therapy including the interleukin 2 antibody basiliximab, and rejection therapy with the anti-CD20 antibody rituximab 4 years earlier. At the time of admission, patient 1 presented with secondary IgG and IgA hypogammaglobulinemia (2.88 g/l IgG, 0.23 g/l IgA and 0.69 g/l IgM) and a maintenance therapy with prednisone, mycophenolate mofetil (MMF), tacrolimus (Fig. 2c). Patient 2 had received a kidney allograft 13 years earlier and

was treated with rituximab for allograft rejection 6 years prior to the SARS-CoV-2 infection. She received immunosuppression with prednisone, azathioprine and tacrolimus (Fig. 2d). Three days prior to SARS-CoV-2 diagnosis, patient 1 was transferred from ward 2 to ward 3 due to neutropenia (Supplementary Fig. 1c) in a room adjacent to that of patient 2. Following this transfer, patient 1 developed a dry cough, low grade fever, and tested positive for SARS-CoV-2 (day 0) (Fig. 2e). Two days later, patient 2 also developed mild respiratory symptoms and was diagnosed with COVID-19 (Fig. 2f). Due to high risk of severe disease, both patients received the REGN-COV mAb cocktail one day post diagnosis (Fig. 2e, f). Patient 1 showed moderate respiratory symptoms requiring oxygen supplementation (Fig. 2g). Hence, MMF was paused and the patient was treated with dexamethasone for 12 days (Fig. 2c). As Patient 2 presented with mild symptoms, immunosuppressive treatment was not modified (Fig. 2d). Due to neutropenia and fever, patient 1 received additional empiric antimicrobial treatment with piperacillin-tazobactam (Supplementary Fig. 1e). Lung scintigraphy excluded pulmonary embolism, but consolidations in line with COVID-19 were detected. Patient 1 showed quick clinical response, i.e. oxygen supplementation could be weaned after one day (Fig. 2g) and the SARS-CoV-2 qPCR cycle threshold (Ct) increased (Fig. 2e). Therefore, immunosuppressive treatment was re-established, followed by discharge at day 16 post COVID-19 diagnosis (Fig. 2a, c). Patient 2 responded well to the antimicrobial treatment and cleared the urogenital infection as indicated by the rapid decrease in C-reactive protein (CRP) (Supplementary Fig. 1f, h). She did not develop severe respiratory symptoms, was discharged two days after COVID-19 diagnosis, and cleared the infection between 6 and 9 days after REGN-COV treatment (Fig. 2f).

Thirty-one days after the initial COVID-19 diagnosis, patient 1 was re-hospitalized due to increased respiratory symptoms, requiring oxygen supplementation (Fig. 2a, g). Unexpectedly, SARS-CoV-2 was diagnosed with a low Ct value of 19, indicating that patient 1 was unable to overcome the infection or was re-infected despite REGN-COV treatment. The Omicron variant BA.1 escaping REGN-COV[23] was excluded by melting curve qPCR analysis. Therefore, reinfection was unlikely. Due to elevated inflammatory parameters, empiric antibiotic treatment with piperacillin/tazobactam was established, and immunosuppressive medication (tacrolimus, MMF and prednisone) was substituted for hydrocortisone (Fig. 2c, Supplementary Fig. 1e). After initial improvement, leading to the reintroduction of the immunosuppression with prednisone and tacrolimus, the patient showed severe respiratory aggravation with hypoxaemia (Fig. 2c). She received high-flow external oxygen (Fig. 2g). Chest radiography identified atypical consolidations, in line with COVID-19 pneumonia. High CRP and procalcitonin values suggested a possible bacterial superinfection, leading to empiric antibiotic therapy with meropenem (Supplementary Fig. 1e, g). *Pseudomonas aeruginosa* was then detected in sputum samples (Supplementary Fig. 1a). Due to IgG deficiency and recurring bacterial and viral infections (Supplementary Fig. 1a), total IgG preparations (Octagam and Intratect) were intravenously administered (IVIG) at day 40 (10 g), 46 (15 g) and 53 (25 g) post COVID-19 diagnosis. Notably, she had not received IVIG prior to day 40. Patient 1 quickly recovered, with SARS-CoV-2 Ct values reaching the detection limit at day 49 and remained high and negative in the following swabs at day 57 and 79, respectively (Fig. 2e). The patient was discharged shortly afterwards (Fig. 2a).

In summary, we describe two immunosuppressed patients infected with the same SARS-CoV-2 Delta AY.43 variant within a nosocomial outbreak. Both were treated with REGN-COV. One patient cleared the virus after a few days. The other patient with secondary hypogammaglobulinemia developed a protracted infection. Clearance of the prolonged SARS-CoV-2 infection was temporally associated with the interruption of the immunosuppressive treatment as well as the substitution with IVIG.

## Full genome sequencing showed an intra-host evolution of SARS-CoV-2

Due to the fact that the patient had not been able to clear the virus, we hypothesized that SARS-CoV-2 adapted to the REGN-COV mAbs. Therefore, we performed full viral genome sequencing and isolated the virus from multiple oropharyngeal swabs over the course of the infection (Fig. 2e). We analyzed the variant frequencies of nucleotide substitutions in comparison to Wuhan-Hu-1 for both the swabs and virus isolates (Fig. 3a, b). Interestingly, we observed four novel high-frequency mutations from day 31 and onward (Fig. 3a). Of these, three resulted in non-synonymous substitutions in the S gene translating to the receptor-binding domain (RBD) mutations K417R, G446V and Y453F (Fig. 3c). The fourth was a non-coding exchange at the three-prime end of the viral genome. Importantly, all mutations initially present at day 0 were detected in subsequent swabs,

demonstrating that the patient was persistently infected with the initial Delta AY.43 variant and not re-infected with a novel virus. We also sequenced the viruses after isolation on Calu-3 cells (passage 1) and confirmed the four novel mutations in the late virus isolates at day 31 and beyond (Fig. 3b). Notably, the virus isolates of day 40 and 43 had an additional N mutation (G96V) and the isolate of day 43 had a two amino acid deletion in the spike gene translating to a deletion of positions 242/243 in the N-terminal domain (NTD) (Fig. 3c). These mutations were already present in lower frequencies in the initial swabs and accumulated to high frequencies in the virus isolates indicating the selection of a subset of viral quasi-species during isolation (Fig. 3a, b). For further analyses, high-titer virus stocks of the d0, d31 and d43 isolates were produced on Calu-3 cells and the genetic stability of this second passage was confirmed by sequencing (Fig. 3b).

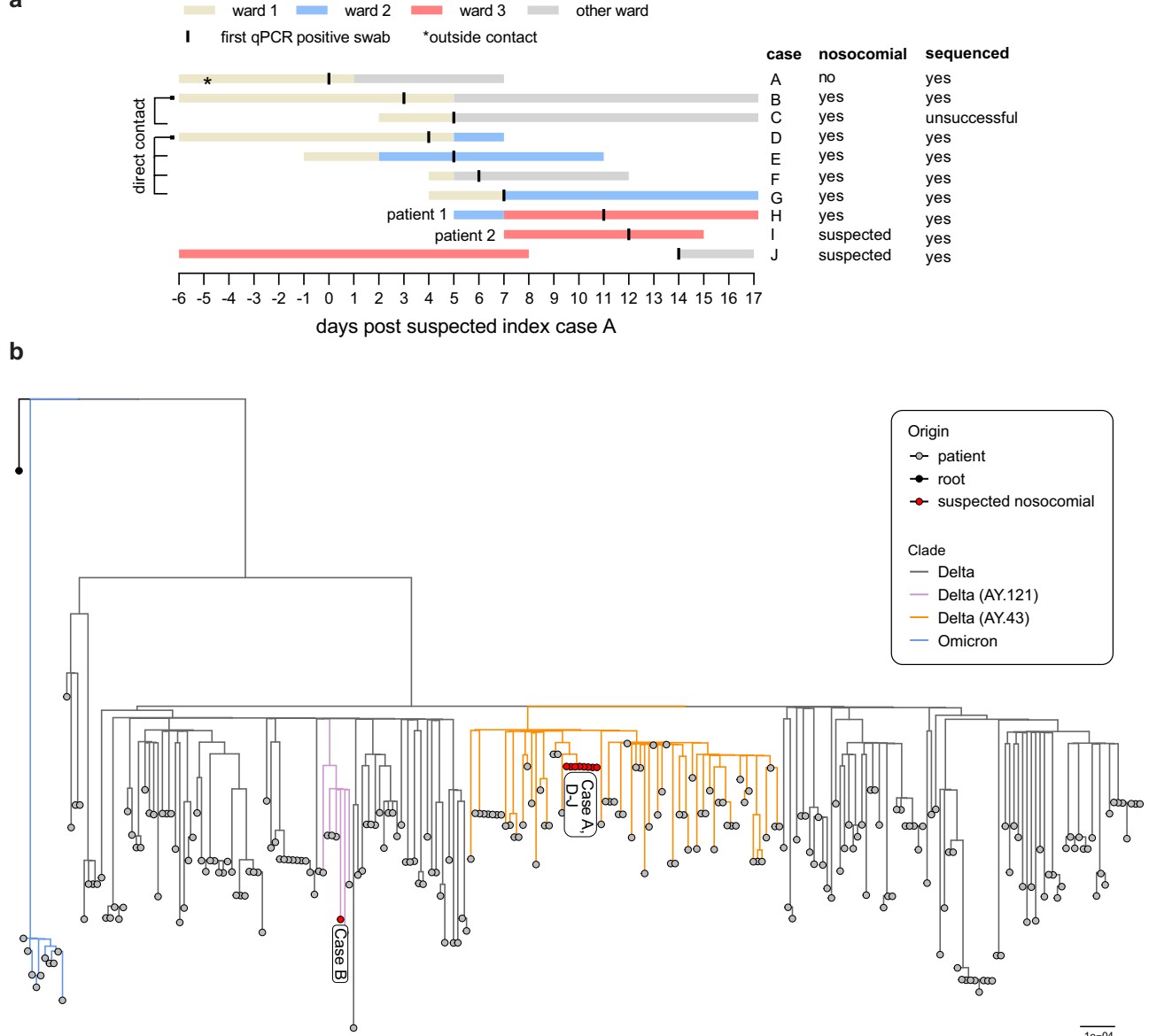

**Fig. 1 | Nosocomial outbreak at the University Medical Centre of Freiburg, Germany caused by Delta lineage AY.43. a** Visualization of known transmission links due to close contacts or spatiotemporal proximity. Extended information is provided in Supplementary Table 1. **b** Phylogenetic tree of all AY.43 sequences generated in Freiburg, Germany, in late 2021 (Supplementary Data 1). The maximum-likelihood phylogenetic tree was constructed with IQ-Tree (1000 bootstrap replicates, GTR + F + R2) and rooted on the Wuhan-Hu-1 reference sequence (NC_045512). The tree was visualized with the R ggtree package. Lineages were assessed with pangolin v0.6 (pangolin data v1.8). Bar indicates substitutions per site.

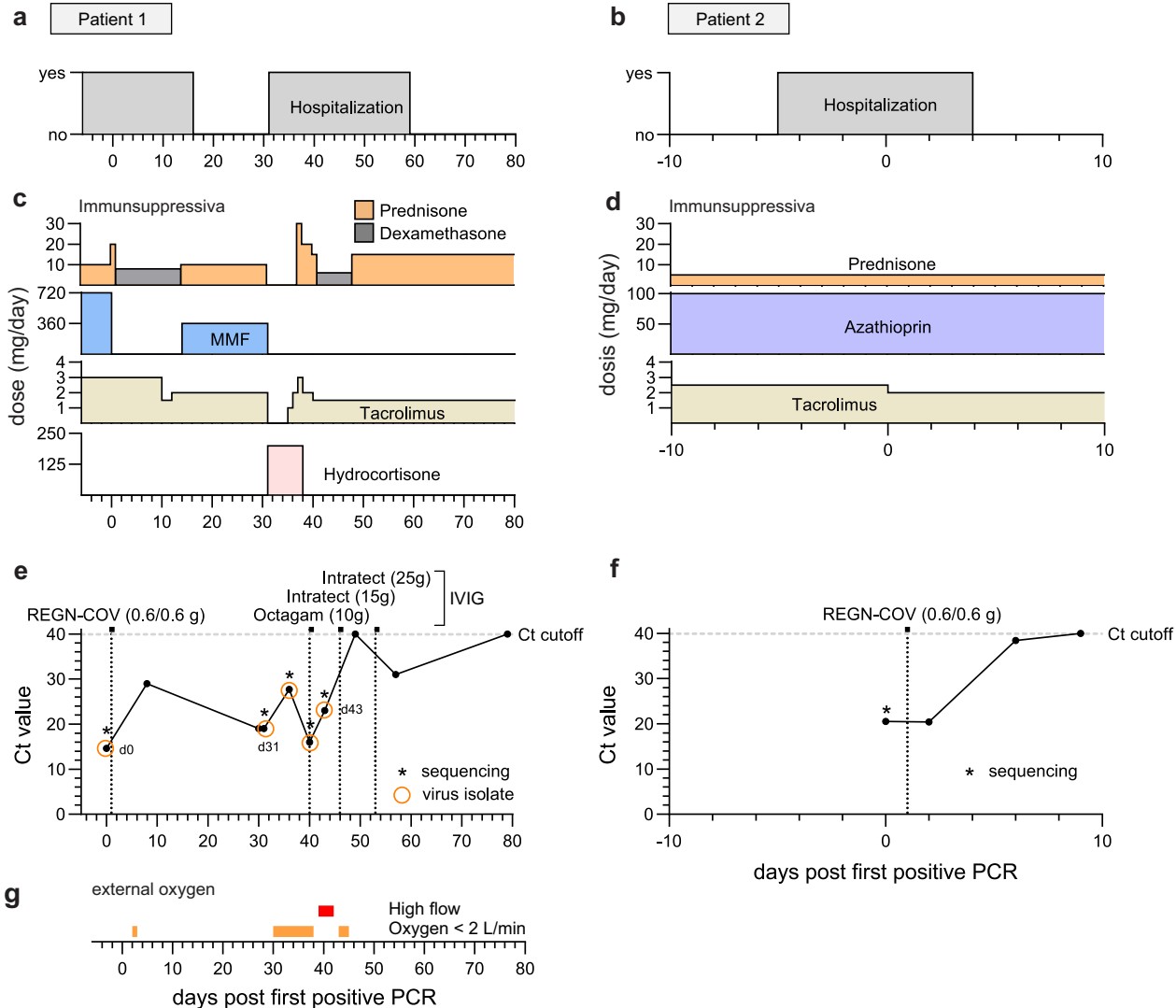

**Fig. 2 | Summary of relevant clinical parameters of the two SARS-CoV-2 infected kidney transplant patients.** Temporal overview of clinical parameters for patient 1 (**a**, **c**, **e**, **g**) and patient 2 (**b**, **d**, **f**). Day 0 indicates the first positive SARS-CoV-2 qPCR result of each patient. **a**, **b** Hospitalization and (**c**, **d**) immunosuppressive drugs of patient 1 and 2. **e**, **f** Diagnostic SARS-CoV-2 qPCR cycle threshold (Ct) values of oropharyngeal swabs. The horizontal dotted lines indicate the cut-off value

(Ct ≥ 40) between positive and negative results. Virus isolation dates are indicated as circles. Vertical dotted lines mark the treatment time points with the mAb cocktail REGN-COV-2 (0.6 g of each Imdevimab and Casirivimab) and total intravenous IgG substitutions (10 g Octagam and 15/25 g Intratect). **g** Need for external oxygen of patient 1.

Next, we evaluated the growth properties of these patient isolates in comparison to the prototypic B.1 isolate Muc-IMB-1 isolated from the first German SARS-CoV-2 patient in 2020[24]. All isolates grew to high titers in both VeroE6 and Calu-3 cells (Fig. 3d, e). The growth of the B.1 isolate outperformed the three patient isolates in both cell lines. Notably, we observed a small growth deficit of one log for the d43 isolate in comparison to the d0 and d31 isolates after 48 h in VeroE6 cells, indicating that the additional N mutation and S deletion might slightly affect viral fitness.

ACE-2 and S binding is mediated by mainly polar interactions via three contact clusters on the RBD. The middle interface consists of positions K417 and Y453, which are two of the three positions where mutations had appeared during viral persistence in patient 1. Notably, the third position, G446, is not involved in binding[25]. Mutation K417R had previously been shown to strengthen salt-bridge interactions in the ACE-2-RBD binding complex as well as RBD-RBD interactions within the trimer of a related Pangolin-CoV[25–27]. Moreover, Y453F that was associated with outbreaks in mink farms in the Netherlands and Denmark[28] was shown to increase binding to mink and human

ACE-2[29,30]. Therefore, we studied possible influences of the S substitutions on ACE-2 binding by surface biolayer interferometry. Despite the published higher affinity to ACE-2 of the individual K417R and Y453F substitutions, we found an approximately tenfold decrease in binding strength (Fig. 3f) and much slower 'on' and 'off' binding kinetics for the S of the d31 compared to that of the d0 isolate (Supplementary Fig. 2a, b). We predicted the S of the d31 isolate with AlphaFold2[31,32] and aligned it to the cryo-electron structure of the S RBD with ACE-2 (Supplementary Fig. 2c). Given that residues 417 and 453 are structurally close to each other and that the phenylalanine at position 453 has been shown to assume more diverse positions in the complex with ACE-2 compared to Y453[30], the combination of these two substitutions, K417R and Y453F, most likely caused the reduced interaction between the viral S and the ACE-2 receptor.

Taken together, SARS-CoV-2 acquired the three S mutations K417R, G446V and Y453F during viral persistence in the REGN-COV treated patient 1. The virus isolates from different time points showed comparable replication capability in cell culture. However, the S protein of the late d31 isolate had a reduced binding capacity to ACE-2.

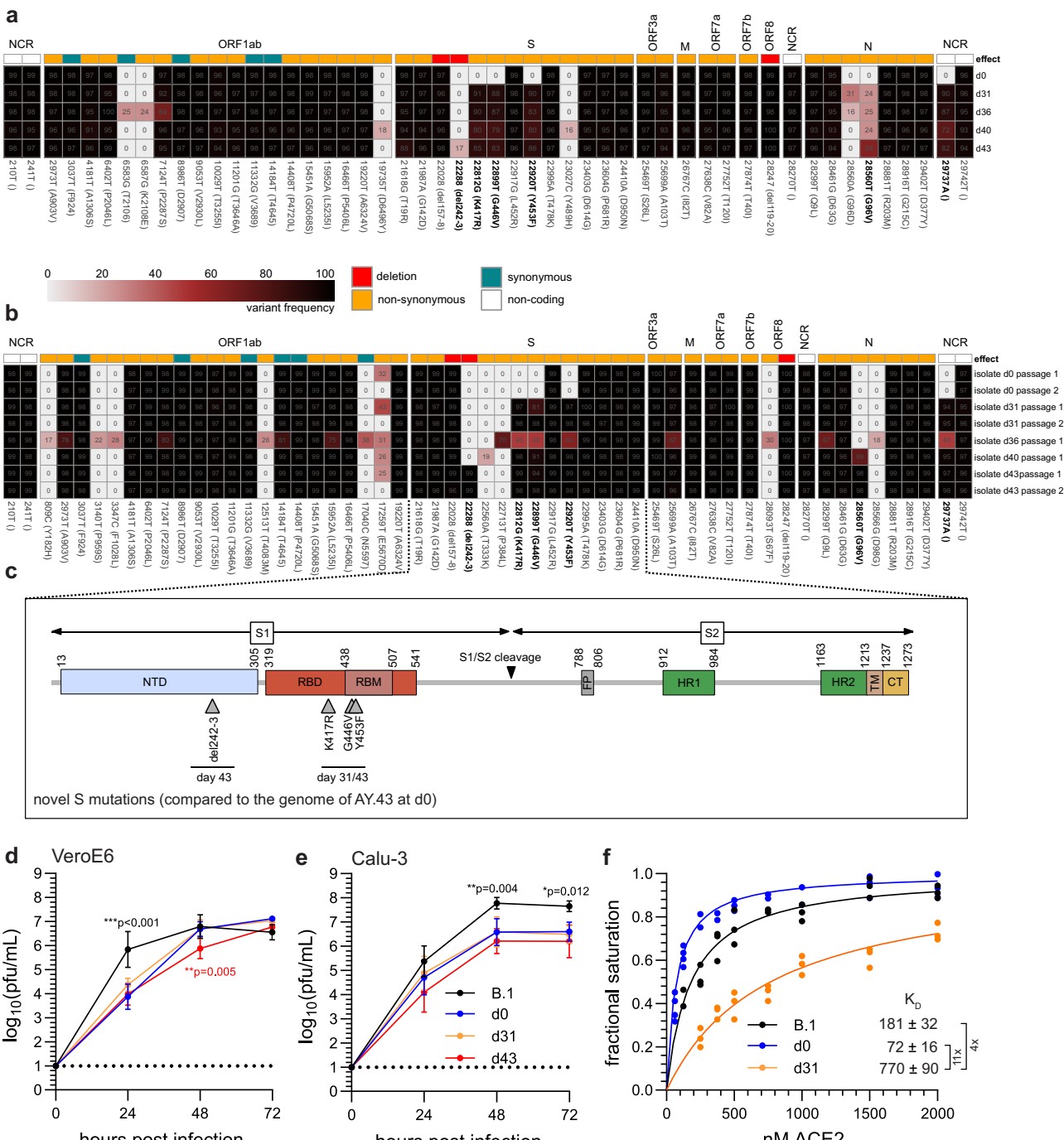

**Fig. 3 | SARS-CoV-2 whole-genome sequencing identifies novel S mutations during viral persistence.** Schematic overview of the viral genome variations of swabs from patient 1 (**a**) and their virus isolates (**b**) in comparison to the Wuhan-Hu-1 reference sequence. The heatmap summarizes the positions in the viral genome and the variant frequencies in the different samples (shown are values above 15%). The days of sampling are indicated at the right, heatmap colour intensity indicates variant frequencies. For the isolates (**b**) sequencing results after virus isolation (passage 1) and additionally for day 0, 31 and 43 sequencing results after virus cultivation (passage 2) are shown. Raw reads are available at ENA (accession number ERP139553). **c** Schematic overview of the SARS-CoV-2 spike protein including the S1 and S2 cleavage products and functional domains such as the N-terminal domain (NTD), receptor-binding domain (RBD), receptor-binding motif (RBM), S1/S2 proteolytic furin cleavage site, fusion peptide (FP), heptad repeat regions (HR1/HR2), transmembrane domain (TM) and C-terminal domain (CT).

Indicated are the novel non-synonymous changes in the spike (S) gene acquired during viral persistence. **d**, **e** Growth of the three patient isolates in **d** VeroE6 and **e** Calu-3 cells. The cells were either infected with a prototypic B.1 isolate (Muc-IMB-1) or one of the patient isolates (d0, d31, d43) using a multiplicity of infection of 0.001. At 24, 48 and 72 h post infection, cell culture supernatants were collected and viral titers were determined by plaque assay. The log-transformed titers are shown as means ± SD of results from four independent experiments. Dotted lines indicate the assay cut-off. Significance was determined via two-way ANOVA with a Sidak´s multiple comparison test (*$p < 0.05$, **$p < 0.01$). **f** Biolayer interferometry binding measurements of ACE-2 association with immobilized S showing variation of fractional signal saturation for different spike proteins. The solid line represents the best fit calculated using Levenberg-Marquardt method. The dissociation constant $K_D$ was calculated from the fitted line as well as from analysis of binding kinetics (Supplementary Fig. 2). Source data are provided in the Source Data file.

## Late isolates potently escape the neutralizing capacity of REGN-COV

REGN-COV is a cocktail of the two mAb Imdevimab and Casirivimab with a high neutralizing capacity against SARS-CoV-2[12,33]. In contrast to previous studies reporting successful viral clearance in immunocompromised individuals[13,34–36] and minimal mutational escape with REGN-COV[22], we observed a persistent SARS-CoV-2 infection and the development of novel mutations after REGN-COV treatment.

To verify that the late virus isolates escaped the neutralizing capacity of REGN-COV, we performed plaque reduction assays (Supplementary Fig. 3a–c) with Imdevimab and Casirivimab and Sotrovimab as a control. Unlike the REGN-COV mAbs, Sotrovimab recognizes an epitope outside of the receptor-binding motif (RBM)[37]. Accordingly, the neutralizing titers 50 ($NT_{50}$) for Sotrovimab were similar for all isolates (Fig. 4a). Imdevimab and Casirivimab potently neutralized the prototypic B.1 and the d0 isolate, but showed significant up to 100-fold decreased $NT_{50}$ values for the d31 and d43 isolates compared to B.1 (Fig. 4b, c). Of note, the viral escape was probably much higher with Imdevimab, as we did not observe plaque reduction at the maximum mAb concentration of 10 µg/ml. (Supplementary Fig. 3b).

Next, we evaluated the in vivo neutralization efficacy of these three antibodies against the three patient isolates in human ACE-2 transgenic mice[38]. Therefore, we intraperitoneally pre-treated the mice with the mAb preparations (50 µg per mouse) 18 h pre intranasal infection with 2.000 pfu of the d0 or d31 isolate (Fig. 4d–g). MAb treated mice infected with the d0 isolate were protected, whereas infected mock treated mice rapidly lost weight and reached humane endpoints between 6 and 8 days post infection (Fig. 4d, e). Analogous to the neutralization data, d31 infected mice were protected by Sotrovimab but not by the REGN-COV mAbs and succumbed to the infection despite the pre-treatment with Imdevimab or Casirivimab (Fig. 4f, g). This demonstrated that the d31 isolate evaded the protective effect of REGN-COV in vivo. The Y453F substitution and multiple substitutions at position 417 had been previously shown to decrease the neutralizing capacity of Casirivimab, whereas mutations at position 446 can facilitate escape from Imdevimab[18]. To structurally analyze the effect of the spike substitutions in the d31 isolate, we aligned the alpha-fold predicted structure of the d31 RBD with the cryo-electron structure of the Wuhan-Hu-1 RBD with the Fab fragments of Imdevimab and Casirivimab[39] (Fig. 4h). This indeed suggested that the Imdevimab escape is likely attributed to the G446V exchange possibly due to hydrophobic changes (Supplementary Fig. 4) and the K417R and Y453F mutations might sterically interfere with the binding between the Fab fragment of Casirivimab and the d31 RBD (Supplementary Fig. 5).

Overall, these data showed the pronounced escape of the d31 isolate from the neutralizing capacity of both Imdevimab and Casirivimab associated with the three S mutations K417R, G446V and Y453F isolated from a persistently infected, REGN-COV treated individual.

## Clearance of the persistent SARS-CoV-2 infection was attributed to the patient's reconstituted immune system

Finally, we aimed to clarify how patient 1 cleared the SARS-CoV-2 infection as this coincided with the supplementation of IVIG as well as the interruption of the immunosuppressive therapy (Fig. 1c, e). As seroprevalence for SARS-CoV-2 is now increasing worldwide, SARS-CoV-2-specific antibodies are likely to be present in IVIG due to convalescent and vaccinated donors, although concentrations largely depend on the manufacturer and the individual batch[40–42]. Unfortunately, we did not gain access to the day 40, 46 and 53 IVIG batches (Fig. 1e) but we were able to test an Intratect batch that the patient received after she had already cleared the SARS-CoV-2 infection. Moreover, we collected blood immediately before and two days after this treatment and evaluated the sera for SARS-CoV-2-specific antibodies using ELISA. Indeed, this Intratect batch showed a low

neutralizing capacity against all four isolates (Supplementary Fig. 6a, b). However, SARS-CoV-2 S-specific IgG in the patient's serum only slightly increased after IVIG substitution (Supplementary Fig. 6b). Based on the assumption that the SARS-CoV-2-specific antibody concentrations in the prior IVIG batches were similar, we hypothesized that the interruption of the immunosuppressive therapy contributed to viral clearance. To investigate this, we compared the serological profiles of all available serum samples from patient 1 and 2. As expected, quantification of total IgG, IgA and IgM showed pronounced hypogammaglobulinemia of patient 1, with IgG concentrations remaining below normal serum levels until the IVIG treatment (Fig. 5a). Interestingly, IgM but not IgA levels increased from day 44, indicating an activation of the patient's adaptive immune response upon the interruption of the immunosuppressive treatment at day 31 (Figs. 5a, 2c). In patient 2, serum concentrations of total IgG, IgA and IgM remained largely unchanged at normal levels (Fig. 5b). This was expected as patient 2 received overall milder immunosuppression compared to patient 1 (Fig. 2c, d). Next, we tested for serum concentrations of SARS-CoV-2-specific antibodies. S-specific IgG vastly increased directly after REGN-COV therapy in both patients followed by a decrease in the first weeks (Fig. 5c, d). However, S-specific IgG was then unchanged in sera of patient 1 between day 30 and 51 (Fig. 5c). This was concurrent with the detection of N-specific IgG for patient 1 but not for patient 2 (Fig. 5e, f). Anti-S/N IgA were below the detection limit in both patients (Fig. 5g, h), but SARS-CoV-2 S/N-specific IgM accumulated in the sera of patient 1 (Fig. 5i, j). We further analyzed anti-S and anti-N IgM separately (Supplementary Fig. 6c, d). Both anti-S and anti-N IgM concentrations markedly increased at day 44, reaching peak concentrations at day 47. Whereas anti-N IgM was only detectable until day 73, anti-S IgM was measurable until day 92. Moreover, anti-S IgM was detectable in higher sera dilutions compared to anti-N IgM in all analyzed samples indicating a more anti-S-based IgM response. The sharp simultaneous increase of SARS-2-CoV-specific IgM and IgG in the sera of patient 1 argued for an activation of a SARS-CoV-2-specific humoral response after day 30. Therefore, we evaluated the variant specific neutralization of the patients' sera against the prototypic B.1 and the three patient isolates (Supplementary Fig. 7). For patient 1, $NT_{50}$ values of the sera at day 10 and 31 showed a high neutralizing capacity against the B.1 and d0 isolate but failed to neutralize the two late isolates (Fig. 5k). Similarly, the sera of patient 2 only neutralized the B.1 and the early d0 isolate (Fig. 5l). In the absence of an immune response, the REGN-COV mAbs likely make up the majority of S-specific IgG in the patients' sera. Therefore, the low neutralizing capacity of the patients' early sera demonstrated again the strong REGN-COV escape of the d31/d43 isolates. Interestingly, sera of patient 1 from day 44 and onward also potently neutralized the late isolates (Fig. 5k). This high neutralizing capacity remained close to the upper limit assay cut-off for the patient isolates and was significantly lower for the B.1 isolate indicating the presence of Delta (AY.43) variant specific antibodies. To further clarify the nature of this humoral response and to exclude that these late neutralizing antibodies originated from the IVIG treatment, we depleted IgG from the sera of patient 1 at day 31, 44, 50 and 82 (Supplementary Fig. 8a) and evaluated the neutralizing effect of IgM still present in the sera (Supplementary Fig. 8b, c). Interestingly, IgG-depleted sera of patient 1 of late time points showed high $NT_{50}$ values for the three patient isolates but only a moderate neutralizing effect against the B.1 isolate (Fig. 5m). This indicates that the patient actively developed highly neutralizing IgM upon interruption of the immunosuppression that was specifically directed against the AY.43 Delta variants still present in this individual.

Although we cannot fully clarify the extent to which the IVIG supported viral clearance, our analysis demonstrated that patient 1 mounted a humoral immune response against SARS-CoV-2, with highly specific and potently neutralizing antibodies against the patient isolates after interruption of the immunosuppressive therapy.

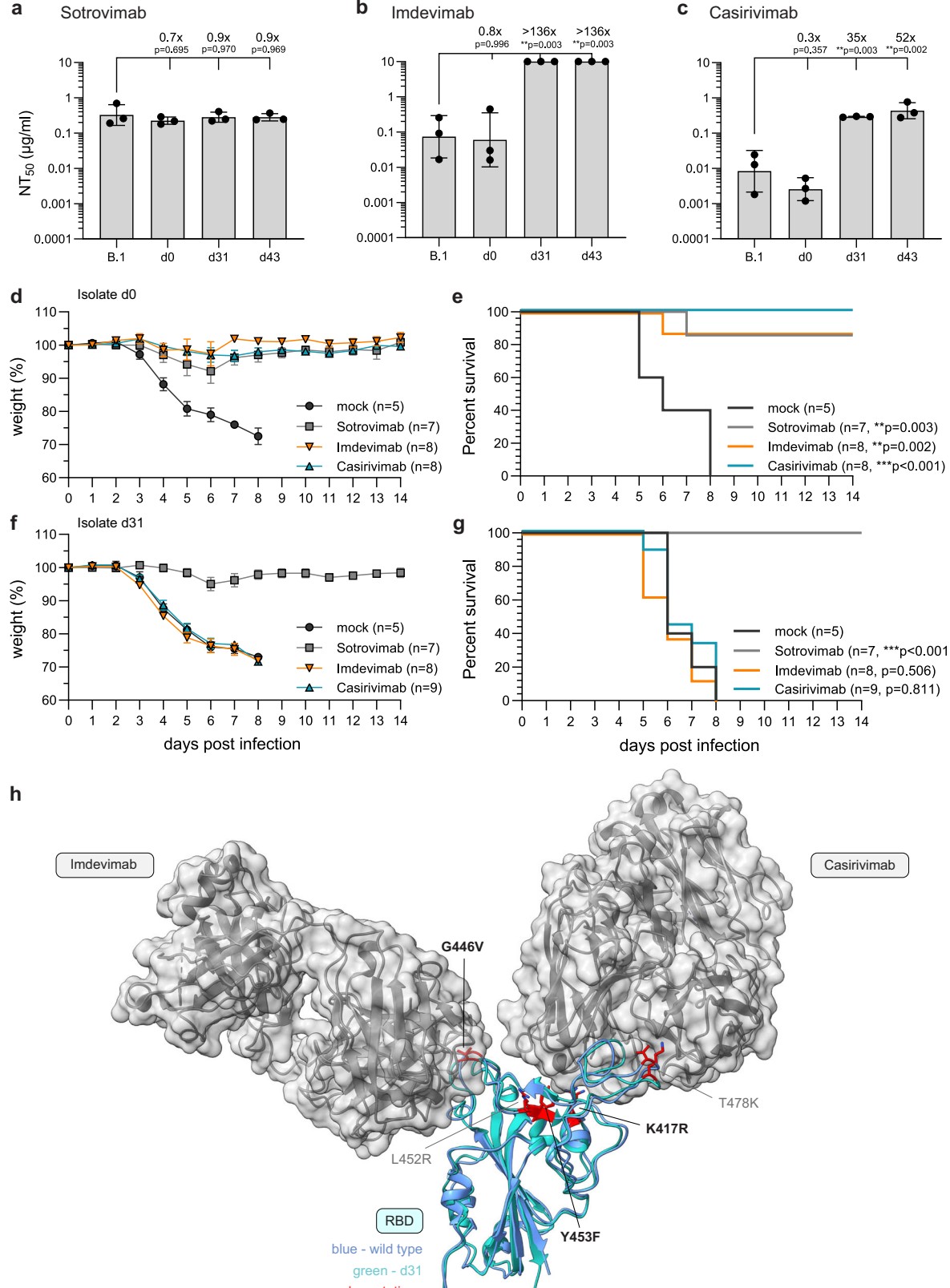

## Discussion

Here, we describe the efficient escape of SARS-CoV-2 from the REGN-COV dual mAb therapy in a severely immunocompromised individual. During the prolonged chronic infection and in the presence of the therapeutic anti-spike antibody treatment, SARS-CoV-2 acquired three S mutations K417R, G446V and Y453F. Each

substitution had already been reported in studies analyzing the potential mutational escape of SARS-CoV-2 from the individual REGN-COV mAbs in cell culture. However, a combination of these mutations is unique and, according to our knowledge, has not been reported to this date[15–18]. Moreover, a study of COVID-19 patients treated with REGN-COV showed that therapy with both of these

**Fig. 4 | REGN-COV antibodies have a reduced neutralizing capacity and fail to protect in vivo against the late isolates of patient 1. a–c** Neutralizing capacity of the therapeutic antibodies **a** Sotrovimab, **b** Imdevimab and **c** Casirivimab. Serial 10-fold dilutions of the monoclonal antibodies were incubated with 100 pfu of the B.1 isolate or the three patient isolates (d0, d31, d43) and analyzed by plaque assay. Neutralization titers 50 ($NT_{50}$) values were calculated from individual curve fits of each serial dilutions ($n = 3$ biologically independent experiments). Shown are geometric mean and geometric standard deviation. Statistics were performed on log-transformed values with a one-way ANOVA (Tukey's multiple comparison test, **$p < 0.01$). **d–g** Weight loss as mean and standard error of means (**d**, **f**) and survival (**e**, **g**) of female, 20–27 weeks old hACE-2 transgenic mice untreated (mock) or treated intraperitoneally with 50 μg/mouse of Sotrovimab, Imdevimab or Casirivimab 18 h pre intranasal infection with 2.000 pfu of the **d**, **e** d0 or **f**, **g** d31 isolate. Age and sex for each animal is provided in the Source Data file. Significance for the survival was calculated with a log-rank (Mantel–Cox) test (**$p < 0.05$, ***$p < 0.001$). **h** 3D presentation of the RBD of the SARS-CoV-2 spike protein (PDB accession number: 6xdg, blue) and the d31 RBD predicted by AlphaFold 2 (green) bound to the Fab fragments of Imdevimab and Casirivimab. RBD mutations compared to Wuhan-Hu-1 (NC_045512.2) are marked in red. Bold amino acid substitutions are only present in the d31 isolate whereas mutations already present in the d0 isolate are marked in grey. Source data are provided in the Source Data file.

non-competing mAbs does not cause mutational escape[21]. Therefore, we hypothesize that this adaptation was only possible under very specific circumstances.

We had the unique opportunity to compare this case with another immunocompromised patient infected with the same SARS-CoV-2 AY.43 variant in the context of a nosocomial infection cluster. Despite having a similar primary disease and receiving immunosuppression, patient 2 responded to her fourth vaccination (738 BAU/ml) and rapidly cleared the infection after REGN-COV treatment (Fig. 5c, d). Seroconversion after vaccination was likely possible due to the milder azathioprine-based triple immunosuppression instead of the more stringent MMF therapy of patient 1, as previously reported[43]. This might have supported viral clearance abreast the REGN-COV therapy and potentially minimized mutational escape due to the presence of additional S-specific antibodies. In contrast, patient 1 presented with a severe secondary hypogammaglobulinemia and neutropenia, both most likely caused by the strong immunosuppressive medication. Moreover, she did not seroconvert after her third SARS-CoV-2 vaccination. After the SARS-CoV-2 infection, patient 1 largely recovered from the initial neutropenia within the first eight days after the infection (Supplementary Fig. 1c) but not from the hypogammaglobulinemia until IVIG treatment and the concurrent reduction of the immunosuppressive medication. We therefore propose that the REGN-COV escape of SARS-CoV-2 in patient 1 was only possible due to selection pressure exerted by the mAbs in the absence of an adaptive immune response that could have disrupted the development of new, mAb-resistant viruses. Importantly, we show that viral clearance only occurred after interruption of the immune therapy and during the development of SARS-CoV-2-specific IgM. IgM not only accounted for a significant proportion of the neutralizing activity of the sera as reported previously[44,45], but also appeared to be highly specific for the Delta AY.43 variant the patient was infected with. Notably, we were not able to assess T-cell-mediated immunity in this retrospective study. However, the cellular immunity in conjunction with the described humoral response likely played an important role. SARS-CoV-2-specific CD4+ and CD8+ T-cell responses are rapidly induced in convalescent and vaccinated individuals and correlate with effective viral clearance[46–48]. Clearance was also concurrent with IVIG substitutions. IVIG can be used for passive immunization reaching pharmacokinetically predictable serum levels[49]. Notably, this is only possible with selected IgG preparations with high concentrations of SARS-CoV-2-specific antibodies. We were not able to test the IVIG the patient received at day 40, 46 and 53, but a later preparation from the same manufacturer only marginally increased blood anti-S concentrations and the neutralizing capacity was moderate against the patient isolates (Supplementary Fig. 6). Although not fully understood, IVIG treatment has immunomodulatory effects including the expansion of regulatory T cells and modulation of dendritic cells (reviewed in ref. 50). Therefore, IVIG could have had indirect effects on the SARS-CoV-2-specific immune response. A concurrent expansion of regulatory T cells might have prevented excessive T-cell activation and inflammation associated with poor clinical outcomes[51].

Due to the presence of variant specific IgM, we concluded that the patient was able to mount an effective adaptive immune response that aided viral clearance. This was concurrent with the reduction of MMF. Previous studies have shown dose-dependent negative effects of MMF on the generation of SARS-CoV-2-specific antibodies in solid organ transplant recipients after vaccination[43,52]. Therefore, we hypothesize that the change in the immunosuppressive treatment regimen facilitated the SARS-CoV-2-specific immune response.

We previously described the evolution of SARS-CoV-2 during viral persistence in another immunosuppressed patient. In this study, the novel mutations did not affect viral fitness but were linked to an immune escape[7]. Similarly, we did not observe an obvious defect in viral fitness of the viruses described here as indicated by the similar growth properties in cell culture and weight loss of infected mice. However, we observed a 10-fold in ACE-2 binding of the d31 S protein with markedly slower 'on' and 'off' binding kinetics. This might be caused by local refolding of the receptor-binding motif of the d31 spike, as the two amino substitutions K417R and Y453F are located in its central part. It is, however, difficult to predict the exact nature of these changes in absence of structural data. Viral evolution is a tight balance between the negative and positive effects of amino acid changes. Allowing substitutions at positions that lower ACE-2 binding might have been one of the few possibilities for SARS-CoV-2 to simultaneously evade both REGN-COV mAbs. Prolonged binding on the other hand might partly compensate this fitness defect and allow for a reasonable efficient viral entry. It is difficult to predict if this virus would be transmissible, but it is possible that due to the ACE-2 affinity reduction, transmission is impaired. Immunocompromised individuals are heavily discussed as a potential source for novel variants that can evade the immune system[53]. Recently, evolution and forward transmission of an Omicron BA.1 sub-lineage with eight additional S mutations that developed in an immunocompromised individual has been observed[54]. The spread of a mAb-resistant SARS-CoV-2 variant in a hospital setting or to the general population can have a dramatic impact on COVID-19 treatment options. This is demonstrated by currently circulating Omicron variants, which can escape multiple therapeutic mAbs[23,55]. Despite patient 1 being part of a nosocomial outbreak, we fortunately have no indication that the here described escape variants were transmitted. We sequenced all SARS-CoV-2-positive patient swabs at the University Medical Centre Freiburg at that time and did not observe a phylogenetically related sequence within the timeframe in which patient 1 was infected (Supplementary Fig. 9). Moreover, no SARS-CoV-2-positive close contact cases were reported during the time the patient was discharged between day 16 and 31.

The case presented here may have important implications for clinicians: (I) Individuals with compromised immune systems may experience escape from combination therapies with mAbs directed against different epitopes, and (II) decrease of the immunosuppression may aid viral clearance thereby impeding forward transmission of a therapy-resistant virus. Collectively, this study underscores the targeted selection pressure of mAbs which can drive a rapid viral evolution within a persistently infected immunocompromised individual.

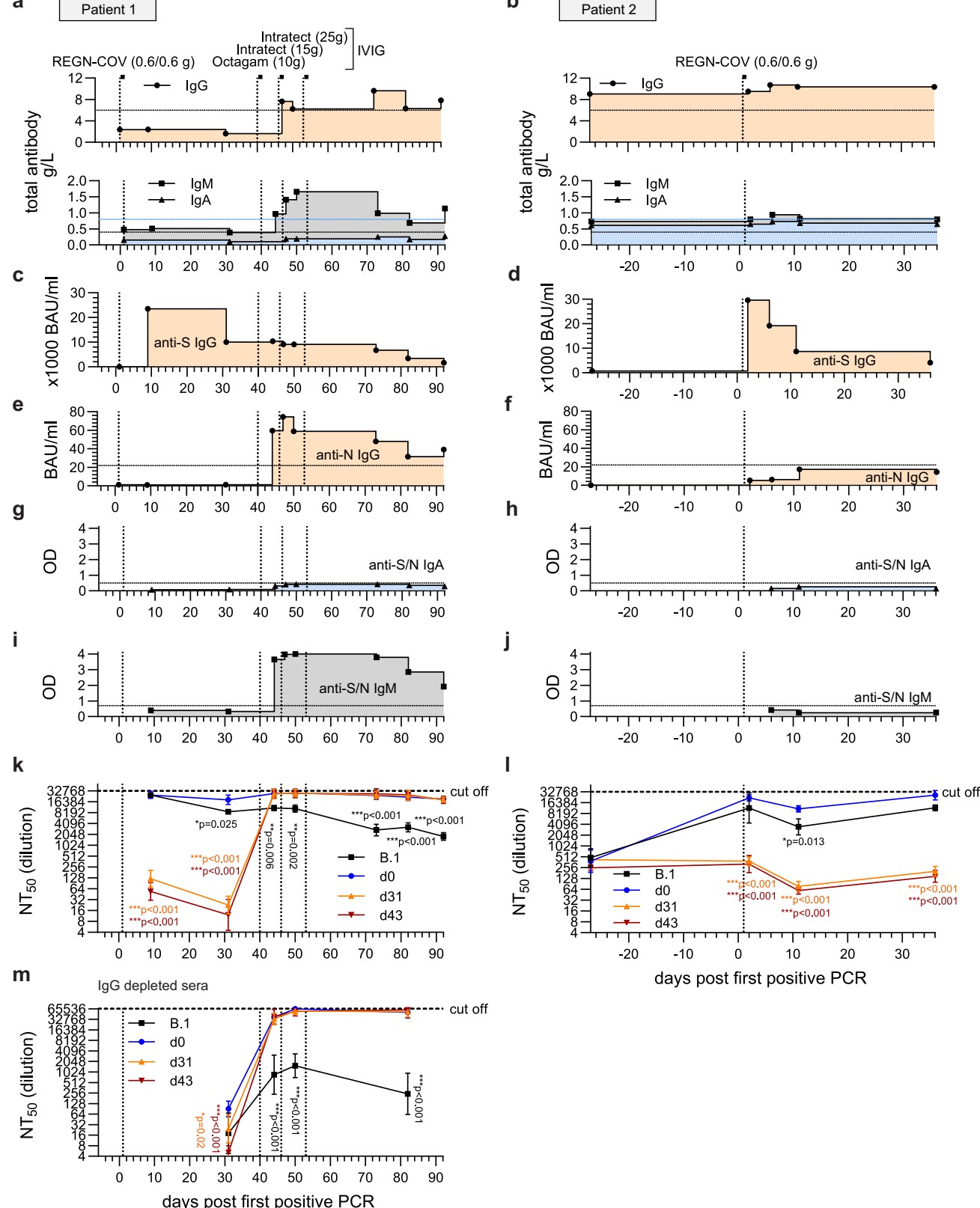

# Methods

## Cell culture

Virus isolation, cell culture and mouse infection experiments with SARS-CoV-2 were performed under Biosafety Level 3 (BSL3) protocols at the Institute of Virology, Freiburg, approved by the Regierungspraesidium Tuebingen (No. 25- 27/8973.10-18 and UNI.FRK.05.16-29).

Adherent African green monkey kidney VeroE6 cells (ATCC CRL-1586) and human lung Calu-3 cells (ATCC HTB-55), kindly provided by Markus Hoffmann (Göttingen), were cultured in 1× Dulbecco's modified Eagle medium (DMEM) containing 5% or 10% foetal calf serum (FCS), respectively. All cell lines were routinely tested for mycoplasma. To isolate SARS-CoV-2 from patient material, filtered throat swabs were

**Fig. 5 | Clearance of the late isolates was associated with decrease in immunosuppression and total IgG substitution.** Temporal overview of serological analyses of patient 1 (**a, c, e, g, i, k, m**) and patient 2 (**b, d, f, h, j, l**). Day 0 indicates the first positive SARS-CoV-2 qPCR result of each patient. Vertical dotted lines mark the treatment time points with the mAb cocktail REGN-COV (0.6 g of each Imdevimab and Casirivimab) and the total intravenous IgG substitutions (10 g Octagam and 15/25 g Intratect). **a, b** Determination of total IgG, IgA and IgM in the patient sera. Detection of **c, d** SARS-CoV-2 S-1-subunit specific IgG, **e, f** SARS-CoV-2 N-specific IgG, **g, h** SARS-CoV-2 S and N-specific IgA and **i, j** SARS-CoV-2 S and N-specific IgM by ELISA. Horizontal lines mark the detection limits.

**k–m** Neutralizing capacity of the sera of **k** patient 1, **l** patient 2 and **m** IgG-depleted sera of patient 1. Serial 2-fold dilutions of the patients' sera were incubated with 100 pfu of the B.1 isolate or the three patient isolates (d0, d31, d43) and analyzed by plaque assay. Plotted are the neutralization titers 50 ($NT_{50}$) values calculated from the individual curve fits of each serial dilutions ($n = 3$ biologically independent experiments). Shown is the geometric mean and geometric standard deviation. Statistics were performed on log-transformed values with a two-way ANOVA (Dunnett's multiple comparison test, $*p < 0.05$, $**p < 0.01$, $***p < 0.001$) and are compared to d0. Source data are provided in the Source Data file.

inoculated on $2 \times 10^6$ Calu-3 cells in 4 ml DMEM with 2% FCS and incubated at 37 °C and 5% $CO_2$ for 4–6 days until the cytopathic effect was visible. The culture supernatant was cleared and stored at −80 °C. Virus titers were determined by plaque assay on VeroE6 cells. Mutations in the viral genomes of the initial isolation and all derived virus stocks were confirmed by next-generation sequencing.

For viral growth kinetics $1 \times 10^6$ VeroE6 or Calu-3 cells were infected with a multiplicity of infection of 0.001 for 1.5 h. Cells were washed three times with PBS and overlaid with 2 mL DMEM with 2% FCS. The supernatants were collected at 24 h, 48 h and 72 h post infection. Viral titers were determined by plaque assay on VeroE6 cells. Besides the patient isolates, the Muc-IMB-1 isolate (lineage B.1) was used as a control (EPI_ISL_406862 Germany/BavPat1/2020)[24], kindly provided by Roman Woelfel, Bundeswehr Institute of Microbiology.

### Virus detection by qPCR
SARS-CoV-2 RNA testing of oropharyngeal swabs was performed using Alinity m SARS-CoV-2 assay (09N78-095, Abbott, Illinois, USA). RNA samples were extracted using the QIAamp MinElute Virus Spin kit (57704, Qiagen, Hilden, Germany). Other viral respiratory pathogens were tested using ePlex RP2 Panels (EA001222, GenMark Diagnostics, La Place Court, United States). CMV DNA was detected in plasma samples using the AltoStar CMV PCR Kit (AS0021513, Altona, Hamburg, Germany). Tests were performed and interpreted according to the manufacturer's instructions and semi-quantitative results reported in cycle threshold (Ct) values.

### Serological testing
Total IgG, IgM and IgA plasma concentrations were determined with the cobas 8000 (Roche Diagnostics). SARS-CoV-2-specific anti-spike protein (S1) IgG (EI2606-9601G, Euroimmun, Medizinische Labordiagnostika AG, Lübeck, Germany) anti-nucleoprotein (N) IgG (7304, Mikrogen Diagnostik GmbH, Neuried, Germany), anti-N/anti-S IgM (ESR400M, Serion, Germany), anti-N IgM (E-EL-E601, Elabscience Biotechnology Inc, USA) and anti-S IgM (LS-F74079, LSBio, USA) ELISAs were performed according to manufacturer's protocol. Results were evaluated semi-quantitatively as arbitrary units (AU) compared to the manufacturer's calibrators or shown as raw values. Neutralizing antibody titers were determined by a plaque reduction assay. Therefore, serial serum or monoclonal antibodies dilutions were incubated with 100 plaque-forming units (pfu) of the SARS-CoV-2 isolates for 1 h. The mixture was dispersed on VeroE6 cells in a 12-well format and cells were overlaid with 0.6% oxoid-agar for 72 h at 37 °C. Fixed cells were stained with 0.1% Crystal violet. The number of plaques was compared with an untreated control without serum or antibodies. To evaluate the neutralizing capacity and determine the neutralizing titer 50 ($NT_{50}$), a non-linear fit least squares regression (constraints: bottom constant equal to 0 and upper constant equal to 100) was performed. To determine the neutralization capacity of SARS-CoV-2-specific IgM in the different sera, IgG was depleted by precipitation with Protein G (Protein G Sepharose 4 Fast Flow) beads. Therefore, beads were centrifuged 1 min at $13,000 \times g$, washed three times with 100 µl PBS and added to the sera. After incubation for 2 h at 4 °C under rotation, beads were pelleted via centrifugation and the supernatant was harvested. To

confirm the successful depletion, IgG and IgM levels before and after depletion were determined with the IgG (Total) Human Uncoated ELISA Kit (88-50550-22) and the IgM Human Uncoated ELISA Kit (88-50620-88, Thermo Fisher, Massachusetts, USA).

### Infection of K18-hACE-2 transgenic mice
Transgenic 034860-B6.Cg-Tg(K18-ACE2)2PrlmanJ mice congenic on the C57BL/6 background were purchased from The Jackson Laboratory and bred locally. Hemizygous 20–27-week-old females were used in accordance to the guidelines of the Federation for Laboratory Animal Science Associations and the National Animal Welfare Body. Mice were housed at 14 h light/10 h dark cycles and temperatures of -18–23 °C with 40–60% humidity. All experiments were performed in compliance with the German animal protection law and approved by the animal welfare committee of the Regierungspraesidium Freiburg (permit G-20/91). Mice were treated intraperitoneal with 50 µg monoclonal antibody (Sotrovimab, Casirivimab, Imdevimab) diluted in 200 µL PBS containing 0.1% BSA. Mice were anaesthetized using isoflurane and infected intranasally with virus diluted in 40 µl PBS containing 0.1% BSA. Mice were monitored daily and euthanized if severe symptoms were observed or bodyweight loss exceeded 25% of the initial weight or was 20% for more than 2 days.

### Protein expression and purification
Full, trimeric spike ectodomains (1-1208) of d0, d31 and the D614G (B.1) spike described before[56], have been cloned into pcDNA.3.1(+) with a set of pre-fusion stabilizing mutations (R682S, R685S, K986P, V987P) and tags described before[57]. All three spike proteins have been expressed exactly as described before for the D614G spike[55]. Briefly, Expi293F cells (Gibco) growing in humidified 8% $CO_2$ atmosphere, at 37 °C, shaking at 125 rpm, in FreeStyle 293 Expression Medium were transfected with 1 mg of spike DNA per 1 L of culture following the manufacturer's instructions. The day after transfection the proteins were moved to 32 °C to increase expression yields[58]. The supernatant was then collected five days after the transfection and protein purified from it with affinity chromatography using TALON Superflow beads. The protein was eluted with PBS plus 200 mM imidazole, concentrated, and gel filtered into 150 mM NaCl, 20 mM TRIS buffer with a Superdex 200 Increase 10/300 GL column (GE Life Sciences). The human ACE-2 ectodomain used in this assay has been exactly expressed as described before[57].

### Biolayer interferometry
Biolayer interferometry measurements of ACE-2 binding to immobilized spikes were performed at 25 °C with 1000 rpm shaking using the Sartorius Octet R8 system. To start with, the spikes at 50–100 µg/mL have been immobilized on NiNTA sensors for 40–60 min. Then ACE-2 association was measured for 2–5 min, followed by dissociation for 10–30 min. Each experiment has been performed at least three times and all results have been included in the analysis. The obtained data have been analyzed with equilibrium and kinetic approaches. The kinetic approach derived the observed rate ($k_{obs}$) from association phases using a single exponential function. This allowed obtaining $k_{on}$ and $k_{off}$ determined from plots of $k_{obs}$ vs ACE-2 concentration. The

equilibrium analysis was performed on data normalized by dividing by the maximum observable response in order to give fractional saturation as a function of ACE-2 concentration.

## Visualization of the spike protein structure

The 3D-structure of the spike RBD was predicted with ColabFold: AlphaFold2 using MMseqs2[59]. The nucleotides 22,493–23,182 of the reference sequence of the SARS-CoV-2 genome (NC_045512.2) were extracted, encoding for the amino acids 311–540 of the spike protein. The following amino acid substitutions were included into the sequence to represent the viral spike protein of day 31: K417R, G446V, L452R, Y453F and T478K. The sequence was inserted in ColabFold as query sequence and the prediction was started with disabled template mode and enabled amber force field. For the calculation of the multi sequence alignment MMseqs2 (UniRef+Environmental) was selected. As model type we selected AlphFold2-ptm with 6 recycle cycles. For the visualization with chimera X of the spike RBD only the amino acids 332–519 were presented. To visualize ACE-2 and S binding the pdb 7u0n and for the binding sites of Imdevimab and Casirivimab the pdb 6xdg was used[39,60]. Both were aligned with the matchmaker tool of Chimera X to the predicted Alpha-fold structure.

## Whole-genome sequencing

For SARS-COV-2 sequencing, the NEBNext ARTIC SARS-CoV-2 FS Library Prep Kit (E7658L, NEB, Frankfurt am Main, Germany) was used. Briefly, cDNA was generated from the RNA of oropharyngeal swabs (57704, QIAamp MinElute Virus Spin kit, Qiagen) or from cell culture supernatants (R1035, Quick-RNA Viral Kit, Zymo Research). The viral genome was then amplified by PCR with primers tiling the entire viral genome. Subsequently, indexed paired-end libraries for Illumina sequencing were prepared. Normalized and pooled sequencing libraries were denatured with 0.2 N NaOH and sequenced on an Illumina MiSeq instrument using the 300-cycle MiSeq Reagent Kit v2 (MS-102-2002, Illumina).

The de-multiplexed raw reads were subjected to a custom Galaxy pipeline, which is based on SARS-CoV-2 analysis pipelines on usegalaxy.eu[61]. The raw reads were pre-processed with fastp[62] and mapped to the SARS-CoV-2 Wuhan-Hu-1 reference genome (Genbank: NC_045512) using BWA-MEM[63]. Primer sequences were trimmed with ivar trim (https://andersen-lab.github.io/ivar/html/manualpage.html). Variants (SNPs and INDELs) were called with the ultrasensitive variant caller LoFreq[64], demanding a minimum base quality of 30 and a coverage of at least 20-fold. Afterwards, the called variants were filtered based on a minimum variant frequency of 10% and on the support of strand bias. The effects of the mutations were automatically annotated in the vcf files with SnpEff[65]. Finally, consensus sequences were constructed by bcftools (v.1.10)[66]. Regions with low coverage or variant frequencies between 0.3 and 0.7 were masked with Ns. Raw sequencing data and deduced consensus genomes have been submitted to the European Nucleotide Archive under the study accession number: ERP139553 [https://www.ebi.ac.uk/ena/browser/view/PRJEB54706]. Final consensus sequences have also been deposited in the GISAID database (www.gisaid.org) (Supplementary Data 1).

An in-house R script was also used to plot the variant frequencies that were detected by LoFreq as a heatmap (pheatmap package v1.0.12) as previously published[7,67]. The script is publicly available (github.com/jonas-fuchs/SARS-CoV-2-analyses, v.1.1 https://doi.org/10.5281/zenodo.7692398) and has also been implemented as a galaxy tool (Variant Frequency Plot on usegalaxy.eu).

## Epidemiological investigation of the nosocomial outbreak

In response to the Delta wave, all patients of the University Medical Centre, Freiburg were screened for SARS-CoV-2 by RT-qPCR at admission and if respiratory symptoms occurred during hospitalization. Nosocomial cases were defined as RT-qPCR negative at admission but positive five days after hospitalization or had direct contact to an infected inpatient. Routine epidemiologic investigation and contact tracing was performed to identify the potential index person and close contact patients or health care workers (HCW). Close contacts were defined According to the contact tracing guidelines published by the Robert Koch Institute, Germany. Moreover, possible transmission links were investigated if COVID-19 diagnosed patients did not share a room but stayed in the same medical ward. All positive samples (Ct < 25) were sequenced by whole-genome sequencing.

For genomic epidemiology, a maximum-likelihood phylogenetic tree was constructed based on SARS-CoV-2 full genome consensus sequences. Therefore, sequences were aligned with MAFFT (v7.45)[68] and a tree was constructed with IQ-Tree (v2.1.2)[69]. The best-fitting substitution model was automatically determined and the tree was calculated with 1000 bootstrap replicates. Branch support was approximated using the Shimodaira–Hasegawa [SH]-aLRT method (1000 replicates). The tree was rooted to the reference sequence NC_045512. The SARS-CoV-2 lineages were classified with pangolin v0.6 (pangolin data v1.8)[70]. To visualize the phylogenetic tree a custom R script was written utilizing the ggtree (v2.2.4)[71], treeio (v1.12.0)[72] and ggplot2 (v3.3.3)[73] packages.

## Plotting and statistical analysis

All plots and statistics were generated with GraphPad Prism v8.4.2 or R studio (R version 4.0.2).

## Ethical statement

The project has been approved by the ethical committee of the Albert-Ludwigs-Universität, Freiburg, Germany. Written informed consent was obtained from all participants and the study was conducted according to federal guidelines, local ethics committee regulations (Albert-Ludwigs-Universität, Freiburg, Germany: No. F-2020-09-03-160428 and no. 322/20) and the Declaration of Helsinki (1975). All routine virological laboratory testing of patient specimens (virus isolation and next-generation sequencing) was performed in the Diagnostic Department of the Institute of Virology, University Medical Center, Freiburg (Local ethics committee no. 1001913).

## Reporting summary

Further information on research design is available in the Nature Portfolio Reporting Summary linked to this article.

# Data availability

The sequence data were submitted to the GISAID database and are publicly available (Supplementary Data 1). The GISAID accession numbers of patient 1 are: EPI_ISL_7996735 (day 0), EPI_ISL_8898226 (day 31), EPI_ISL_8898236 (day 36), EPI_ISL_9324089 (day 40) and EPI_ISL_9324138 (day 43). Raw sequencing data and deduced consensus genomes have been also submitted to the European Nucleotide Archive under the study accession number: ERP139553. The pdb database accession numbers [https://www.rcsb.org/] that were used for structural analysis and visualization are: 7U0N (doi:10.2210/pdb7U0N/pdb, SARS-CoV-2 S binding ACE2) and 6XDG (doi: 10.2210/pdb6XDG/pdb, REGN-COV binding SARS-CoV-2 S). For SARS-CoV-2 lineage assignment pangolin data v1.8 [https://github.com/cov-lineages/pangolin-data/releases/tag/v1.8] was used. The sequence used in this study as the SARS-CoV-2 reference genome has the GenBank [https://www.ncbi.nlm.nih.gov/genbank/] accession: NC_045512 (Wuhan-Hu-1). All other data are available in the article and its Supplementary files or from the corresponding author upon request. Source data are provided with this paper.

# Code availability

The script to visualize the variant frequencies is publicly available (github.com/jonas-fuchs/SARS-CoV-2-analyses, v.1.1, https://zenodo.

org/badge/latestdoi/336032336) and implemented on usegalaxy.eu (Variant Frequency Plot).

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

## Acknowledgements

We thank Roman Woelfel (Bundeswehr Institute of Microbiology) for providing the B.1 (Muc-IMB-1) isolate and Markus Hoffmann (Goettingen) for the Calu-3 cells. We are grateful to Otto Haller for helpful comments on the manuscript. This work was funded by the Deutsche Forschungsgemeinschaft (DFG, German Research Foundation, grant number PA 2274/4-1) and by the Bundesministerium fuer Bildung und Forschung (BMBF) through the Deutsches Zentrum fuer Luft- und Raumfahrt, Germany to M.P. and M.S. (DLR, grant number 01KI2077). T.W. and E.N.H. were also supported by the DFG (Grant No. 466417053 (to T.W.) and Project-ID 431984000–CRC 1453 Nephrogenetics (to E.N.H. and T.W.)). A.G.W., P.N., S.R.M., and S.J.G. were supported by the Francis Crick Institute which receives its core funding from Cancer Research UK (FC001078), the Medical Research Council (FC001078) and the Wellcome Trust (FC001078). The funders had no role in the study design, data analysis, data interpretation and in the writing of this report. All authors had full access to the data in the study and accept responsibility to submit for publication.

## Author contributions

J.F., M.P., L.J., S.W., L.K., D.H. and D.S. designed the study and contributed to experiment design and data interpretation. E.N.H., K.W., M.P., D.H., T.D. and T.W. performed clinical management, sample collection and evaluation and analysis of clinical data. S.K., S.R. and J.F. performed genomic epidemiological analyses. L.J. and J.F. performed bioinformatic analyses. S.W. did the prediction and visualization of the SARS-CoV-2 spike. L.J., L.K., S.W., A.G.W., A.B.G., J.B., M.D., P.K., L.G., P.N., G.K., S.R.M. and J.F. performed experiments and analyzed and processed the data. M.P., G.K., S.J.G., T.W., E.N.H. and MS were involved in funding acquisition. J.F. and L.J. wrote the manuscript.

## Funding

## Competing interests

The authors declare no competing interests.
