## [Peer Review File · Nature Communications]

Total escape of SARS-CoV-2 from dual monoclonal antibody therapy in an immunocompromised patientREVIEWER COMMENTS

Reviewer #1 (Remarks to the Author):

As denoted by the title, the authors report a case of immune escape to a monoclonal antibody cocktail (dual mAb). The manuscript is well written. The investigation is comprehensive and the findings are articulated clearly. The findings echo the collective experience of monoclonal antibodies being rendered ineffective by variants and subvariants of SARS-CoV-2 (in this case SARS146 CoV-2 Delta AY.43 where the combination of 3 mutations was thought to be responsible for the escape). My suggestions are minor/editorial in nature.

Minor

“corona-virus” change to coronavirus

“CRP and procalcitonin values also indicated a possible bacterial superinfection” might change to “CRP and procalcitonin values suggested a possible bacterial superinfection”

“etiopathology” unclear what this means

“we performed a comparative serology” suggest something to the effect of “we compared the serological profiles”

“can aid viral clearance thereby prohibiting forward transmission” suggest substituting “prohibiting” with something else e.g. “impeding”

Figure 2 legend: spelling “circels” should be “circles”

Reviewer #2 (Remarks to the Author):

This article is a well-written thorough description of a nosocomial COVID-19 outbreak and tracks the clinical response of two specific patients, both of which are immunocompromised, in different ways. The antibody response is thoroughly characterized and escape virus variants are identified in one of the patients.

Major Comments:

1. For Patient 1, is there any way to distinguish anti-S/N IgM and separate these? Was the IgM response more Spike or N protein based? This is an important distinction since this is the main hypothesis for viral clearance in this patient.
2. Line 335- for the d31 virus which showed a 10-fold decrease in ACE2 binding, do you have any hypotheses as to which specific mutations are contributing to this? Both in the results and discussion there is not much discussion as to the specific mutations identified and their contributions.
3. Paragraph at line 356: I recommend adding a third implication for clinicians along the lines of the specific combination of neutropenia + hypogammaglobulinemia and that these patients are at especially high risk, or If either of these are due to induction therapy that could be an important point of discussion.
4. Discussion – I recommend adding a more in depth discussion about the effects of neutropenia and hypogammaglobulinemia in the immune response for Patient 1, how much do you think each one of these contributed to the immune response? As well as how recently these patients had renal transplant and induction therapy which can also significantly alter immune responses.

Reviewer #3 (Remarks to the Author):

Authors for this manuscript described the identification of one case of Spike escaping mutations in an immunocompromised person under REGN-COV (combination of 2 mAbs) treatment. The authors further showed that the clearance of the mutated virus was likely due to de novo development of IgM after the disruption of immunosuppressants. The authors concluded that mAb treatment in the absence of aided or de novo polyclonal response may result in selection of mutants that are resistant to multiple mAbs, and proposed that decreasing immunosuppression could aid virus clearance and prevent selection of drug resistant variants.

The case is well presented. The escaping mutations are well characterized both in vitro and in vivo. The potential contribution of IgM response in virus clearance is also supported by experimental data. The study provides important findings that could affect decisions on treating of COVID-19 infections in immunocompromised population.

Minor comments for authors:

Line 129: "Omicron variant BA.1 escaping REGN-COV was excluded by melting curve qPCR analysis[23]". Ref 23 should be moved to after escaping REGN-COV since it is a reference for the neutralization escape from mAbs, not a reference for method of detection?

Figure 3f: Any reason for missing D43 isolate's RBD in the test?

Line 278: "This high neutralizing capacity remained close to the assay cut off for the patient isolates and was significantly lower for the B.1 isolate..." Would be helpful to add "(upper limit) before "assay cut off".

POINT-BY-POINT RESPONSE TO THE REVIEWERS

We would like to thank all three reviewers for the thorough revision of our manuscript and the helpful comments and suggestions. Please find in the following the point-by-point response to your critical evaluation. Our responses to your points is marked in green.

REVIEWER COMMENTS

Reviewer #1 (Remarks to the Author):

As denoted by the title, the authors report a case of immune escape to a monoclonal antibody cocktail (dual mAb). The manuscript is well written. The investigation is comprehensive and the findings are articulated clearly. The findings echo the collective experience of monoclonal antibodies being rendered ineffective by variants and subvariants of SARS-CoV-2 (in this case SARS146 CoV-2 Delta AY.43 where the combination of 3 mutations was thought to be responsible for the escape). My suggestions are minor/editorial in nature.

Minor

“corona-virus” change to coronavirus

We changed this in line 50.

“CRP and procalcitonin values also indicated a possible bacterial superinfection” might change to “CRP and procalcitonin values suggested a possible bacterial superinfection”

We changed this to “*CRP and procalcitonin values suggested a possible bacterial superinfection*” in line 140/141.

“etiopathology” unclear what this means

We changed this to “*Due to the fact that the patient had not been able to clear the virus*” in line 156.

“we performed a comparative serology” suggest something to the effect of “we compared the serological profiles”

We changed this to “*we compared the serological profiles*” in line 258.

“can aid viral clearance thereby prohibiting forward transmission” suggest substituting “prohibiting” with something else e.g. “impeding”

We changed “prohibiting” to “*impeding*” in line 375.

Figure 2 legend: spelling “circels” should be “circles”

We corrected this in line 762.

Reviewer #2 (Remarks to the Author):

This article is a well-written thorough description of a nosocomial COVID-19 outbreak and tracks the clinical response of two specific patients, both of which are immunocompromised, in different ways. The antibody response is thoroughly characterized and escape virus variants are identified in one of the patients.

Major Comments:

1. For Patient 1, is there any way to distinguish anti-S/N IgM and separate these? Was the IgM response more Spike or N protein based? This is an important distinction since this is the main hypothesis for viral clearance in this patient.

We thank the reviewer for this helpful question. As the anti-N IgG response is rather low and IgM is highly neutralizing, we hypothesise that the IgM response is largely directed against the spike protein. To test this hypothesis, we assessed the response in separate anti-S and anti-N IgM ELISAs. Indeed, we found both anti-S and anti-N IgM, but anti-S IgM was detectable in higher sera dilutions and in later serum samples indicating a more anti-S based response. We now mention this in line 272-277 and show the data in Suppl. Figure 6 c/d.

2. Line 335- for the d31 virus which showed a 10-fold decrease in ACE2 binding, do you have any hypotheses as to which specific mutations are contributing to this? Both in the results and discussion there is not much discussion as to the specific mutations identified and their contributions.

We have partially addressed this question in lines 193-197, where we introduced the results of the binding assays. We explain, that residues 453 and 417 of the spike RBD are closely located to each other and that previous structural data suggested that 453 can assume diverse conformations. Hence, we speculate that the unique combination of the mutations 417R and 453F likely changed the local structure of this region in the d31 spike thus resulting in its weaker binding to the receptor.

Following the reviewers' helpful comments, we have now discussed this in more detail (line 349-352).

3. Paragraph at line 356: I recommend adding a third implication for clinicians along the lines of the specific combination of neutropenia + hypogammaglobulinemia and that these patients are at especially high risk, or if either of these are due to induction therapy that could be an important point of discussion.

We would like to thank the reviewer for this suggestion. Please see the answer to the next point (point 4) as both points are closely connected.

4. Discussion – I recommend adding a more in depth discussion about the effects of neutropenia and hypogammaglobulinemia in the immune response for Patient 1, how much do you think each one of these contributed to the immune response? As well as how recently these patients had renal transplant and induction therapy which can also significantly alter immune responses.

Patient 1 and 2 received their kidney transplantations 4 and 13 years prior to their SARS-CoV-2 infection, respectively. We have now clarified the history of transplantation and induction therapy in more detail (line 101-103 and line 105-107). As the induction therapy was four years in the past, we

assume that this did not cause the observed neutropenia and hypogammaglobulinemia of patient 1. We rather think that the maintenance of the strong triple immune suppression of patient 1 was the most likely cause. We have clarified this in the discussion in line 323/325.

The neutropenia of patient 1 is only pronounced prior to and in the first 8 days after the SARS-CoV-2 infection (Supplementary figure 1c). During this time period, the patient did not develop severe symptoms. Therefore, we postulate that the recovery from the hypogammaglobulinemia due to the reduction of the immunosuppressive therapy and IVIG treatment is the major contributor to viral clearance and stabilization of the overall medical condition. We now discuss this in line 326-329.

Neutrophils have been shown to be elevated in severe COVID-19 patients and thus neutrophilia has been described in several studies as an indicator of poor COVID-19 outcomes. Hypogammaglobulinemia has been found to be predictive for severe COVID-19. As such, both predictors are counter indicative for the severity. As we discuss one rather complex clinical case where we also observed multiple bacterial and viral infections (Supplementary figure 1a) on top of SARS-CoV-2 we do not feel comfortable to suggest and discuss that the combination of neutropenia and hypogammaglobulinemia is a predictor for poor COVID-19 outcomes.

Reviewer #3 (Remarks to the Author):

Authors for this manuscript described the identification of one case of Spike escaping mutations in an immunocompromised person under REGN-COV (combination of 2 mAbs) treatment. The authors further showed that the clearance of the mutated virus was likely due to de novo development of IgM after the disruption of immunosuppressants. The authors concluded that mAb treatment in the absence of aided or de novo polyclonal response may result in selection of mutants that are resistant to multiple mAbs, and proposed that decreasing immunosuppression could aid virus clearance and prevent selection of drug resistant variants.

The case is well presented. The escaping mutations are well characterized both in vitro and in vivo. The potential contribution of IgM response in virus clearance is also supported by experimental data. The study provides important findings that could affect decisions on treating of COVID-19 infections in immunocompromised population.

Minor comments for authors:

Line 129: "Omicron variant BA.1 escaping REGN-COV was excluded by melting curve qPCR analysis[23]". Ref 23 should be moved to after escaping REGN-COV since it is a reference for the neutralization escape from mAbs, not a reference for method of detection?

Thank you, the citation was indeed misplaced. We have moved it as suggested (line 133).

Figure 3f: Any reason for missing D43 isolate's RBD in the test?

We appreciate this question. The main difference between the day 43 and day 30 spikes is the deletion of position 242-243. This deletion is only present in low frequencies in the patient swab (17 %, Fig.3 a) and was selected during virus isolation. Moreover, this difference is located in the NTD which is not involved in receptor binding. We have not included this as this is not a main difference in the original patient material and likely does not impact ACE2 binding.

Line 278: “This high neutralizing capacity remained close to the assay cut off for the patient isolates and was significantly lower for the B.1 isolate...” Would be helpful to add “(upper limit) before “assay cut off”.

As suggested we added “upper limit assay cut off” in line 289.

REVIEWER COMMENTS

Reviewer #1 transfusion therapy (Remarks to the Author):

The prior comments have been adequately addressed.

Reviewer #2 antibody, SARS-CoV-2 (Remarks to the Author):

No new comments

Reviewer #3 SARS-CoV-2 (Remarks to the Author):

The authors have addressed my suggestions/questions adequately.

Reviewer #4 solid organ transplantation (Remarks to the Author):

The concern of failure to eradicate viral infection/shedding and development viral mutations has been a concern for transplant professionals. The authors report nosocomial SARS-CoV-2 spread in kidney recipients. Both recipient were years out and on stable immunosuppressive regimens. Both received combination anti-SARS-CoV-2 monoclonal therapy. One KT recipient (pt 2 with UTI) cleared the viral infection promptly and the other (pt 1 with acquired hypoglobulinemia) had a protracted infection/disease with selection/emergence of clinical virus with spike protein variants with lesser binding to therapeutic antiviral prep. The persistent infection (>month) eventually cleared virus with IVIg and reduction in immunosuppression. The authors demonstrated that antibody activity persisted in the serum via neutralization assays of control virus.

Immunoglobulins (mostly IgG, but occ IgM) have had a long use in transplantation for modulation of B and T cell activity (Sivasi, Clin Exp Immunol 2000, attached) in addition to their activity in prophylaxis/treatment of viral infections (CMV, parvovirus, BKV and HBV). The specific cellular mechanisms for efficacy are not worked out, but IVIG appears to affect multiple T cell activities. Additionally, viral infection and vaccine elicit not only a B cell responses, but T cell irrespective of generated ab (Sattler, JCI 2021). The interplay between soluble and cellular immunity is complex. A few questions

1. The amount and frequency of IVIG given to patient 1 is not mentioned. This would be useful information. It was also mentioned that the pt. had secondary hypogammaglobulinemia...how low/hypo- was she? Had she received IVIG previously for suboptimal amt of IgG?

While txp center protocols vary, clinical protocols intending to modify cellular responses (antibody mediated rejection or alloantibody risk) are typically in the range of 500-1000 ml/kg/dose for 3-5 administrations.

Were any assessments of T-cell response performed? Either non-specific T cell responses or specific to SARS-CoV-2 antigens.

2. Immunosuppression modulation: Inclusion of drugs/doses for at least patient 1 on the timeline in Fig 1 would be helpful. The authors state that modification of immunosuppression was likely important for viral clearance. However, there were multiple clinical events (time/exposure to virus, additional IVIG and chemotherapeutic modulation) that may/not be relevant to viral clearance and recovery. Modulation of antimetabolites that impact intracellular T-cell purine metabolites is prob important, but conjectural. Transplant recipients remain PCR+ on NP swabs for longer periods of time than non-IS people (about 1 month being average, but 3-5 months not being unusual).

The authors have demonstrated that protracted viral replication/shedding in the presence of anti-spike antibodies was associated with emergence of virions spike proteins that had lesser affinity to the antibody.

Reviewer #4 also attached two papers which are cited below:

Sivasai KS, Mohanakumar T, Phelan D, Martin S, Anstey ME, Brennan DC. Cytomegalovirus immune globulin intravenous (human) administration modulates immune response to alloantigens in sensitized renal transplant candidates. *Clin Exp Immunol*. 2000 Mar;119(3):559-65. doi: 10.1046/j.1365-2249.2000.01138.x. PMID: 10691931; PMCID: PMC1905589.

Sahin U, Muik A, Derhovanessian E, Vogler I, Kranz LM, Vormehr M, Baum A, Pascal K, Quandt J, Maurus D, Brachtendorf S, Lörks V, Sikorski J, Hilker R, Becker D, Eller AK, Grützner J, Boesler C, Rosenbaum C, Kühnle MC, Luxemburger U, Kemmer-Brück A, Langer D, Bexon M, Bolte S, Karikó K, Palanche T, Fischer B, Schultz A, Shi PY, Fontes-Garfias C, Perez JL, Swanson KA, Loschko J, Scully IL, Cutler M, Kalina W, Kyratsous CA, Cooper D, Dormitzer PR, Jansen KU, Türeci Ö. COVID-19 vaccine BNT162b1 elicits human antibody and TH1 T cell responses. *Nature*. 2020 Oct;586(7830):594-599. doi: 10.1038/s41586-020-2814-7. Epub 2020 Sep 30. Erratum in: *Nature*. 2021 Feb;590(7844):E17. PMID: 32998157.

POINT-BY-POINT RESPONSE TO THE REVIEWERS

We would like to thank all four reviewers for the thorough revision of our manuscript and for the helpful comments and suggestions. Please find in the following the point-by-point response to the critical evaluation of reviewer 4. Our responses are marked in green.

REVIEWER COMMENTS

Reviewer #1 transfusion therapy (Remarks to the Author):

The prior comments have been adequately addressed.

Reviewer #2 antibody, SARS-CoV-2 (Remarks to the Author):

No new comments

Reviewer #3 SARS-CoV-2 (Remarks to the Author):

The authors have addressed my suggestions/questions adequately.

Reviewer #4 solid organ transplantation (Remarks to the Author):

The concern of failure to eradicate viral infection/shedding and development viral mutations has been a concern for transplant professionals. The authors report nosocomial SARS-CoV-2 spread in kidney recipients. Both recipient were years out and on stable immunosuppressive regimens. Both received combination anti-SARS-CoV-2 monoclonal therapy. One KT recipient (pt 2 with UTI) cleared the viral infection promptly and the other (pt 1 with acquired hypoglobulinemia) had a protracted infection/disease with selection/emergence of clinical virus with spike protein variants with lesser binding to therapeutic antiviral prep. The persistent infection (>month) eventually cleared virus with IVIg and reduction in immunosuppression. The authors demonstrated that antibody activity persisted in the serum via neutralization assays of control virus.

Immunoglobulins (mostly IgG, but occ IgM) have had a long use in transplantation for modulation of B and T cell activity (Sivasi, Clin Exp Immunol 2000, attached) in addition to their activity in prophylaxis/treatment of viral infections (CMV, parvovirus, BKV and HBV). The specific cellular mechanisms for efficacy are not worked out, but IVIG appears to affect multiple T cell activities. Additionally, viral infection and vaccine elicit not only a B cell responses, but T cell irrespective of generated ab (Sattler, JCI 2021). The interplay between soluble and cellular immunity is complex. A few questions

The reviewer raises an important point. We very much appreciate this comment. To clarify this, we did not intend to state that the highly neutralizing IgM were the sole reason why the patient cleared the virus. We also think, that cellular immunity likely played an important part. As this was a retrospective study, no heparin blood or PBMCs were bio-banked. Unfortunately, we therefore had no material to analyse the cellular responses. The SARS-CoV-2 specific neutralisation measurements was used as a surrogate to show a pathogen-specific immune response (now stated in line 352/353). We now also discuss the role of T cells for viral clearance (line 337-341) and the possible role of IVIG for the immune modulation (line 346-351).

1. The amount and frequency of IVIG given to patient 1 is not mentioned. This would be useful information.

Until day 80, patient 1 had received IVIG at day 40, 46 and 53 (line 145). The amount of IVIG given at the respective timepoints are shown in figure 2e and figure 5a (10, 15, 25 g). To further clarify this, we have added “IVIG” to both figures and added the amounts to the text (line 145/146).

It was also mentioned that the pt. had secondary hypogammaglobulinemia... how low/hypo- was she?

We measured the amount of total IgG, IgM and IgA for both patients from the time of infection and onward (shown in Figure 5 a/b). We have also added to the text the measurement for patient 1 at the time of admission (line 104/105). She had reduced levels of total IgG and IgA, but normal levels of total IgM. All levels of these measurements at the time of admission were also in a comparable range to the measurements at day 0.

Had she received IVIG previously for suboptimal amt of IgG? While txp center protocols vary, clinical protocols intending to modify cellular responses (antibody mediated rejection or alloantibody risk) are typically in the range of 500-1000 ml/kg/dose for 3-5 administrations.

We thank the reviewer for this question. The clinicians discussed prior IVIG treatment, but she was not treated before day 40. However, IVIG treatment continued after day 80. We now mention this in line 146.

Were any assessments of T-cell response performed? Either non-specific T cell responses or specific to SARS-CoV-2 antigens.

As stated above, we retrospectively analysed the patient and no heparin blood or PBMCs were bio-banked. Therefore, we were not able to assess T cell responses. We now mention this in the discussion (line 337-341).

2. Immunosuppression modulation: Inclusion of drugs/doses for at least patient 1 on the timeline in Fig 1 would be helpful.

We thank the reviewer for this suggestion. We would not like to add the dose to figure 1, however, we depicted the drugs and respective treatment regimens for both patients in figure 2 c and d. We have now changed the y-axis label from “mg/day” to “dose (mg/day)” to clarify this further.

The authors state that modification of immunosuppression was likely important for viral clearance. However, there were multiple clinical events (time/exposure to virus, additional IVIG and chemotherapeutic modulation) that may/not be relevant to viral clearance and recovery. Modulation of antimetabolites that impact intracellular T-cell purine metabolites is prob important, but conjectural. Transplant recipients remain PCR+ on NP swabs for longer periods of time than non-IS people (about 1 month being average, but 3-5 months not being unusual).

We fully concur with this notion. Indeed, this clinical case was highly complex and viral clearance coincided with multiple events. Given the data, our best hypothesis is that a reduction in immunosuppressive medication allowed for a SARS-CoV-2 specific immune response to be mounted. We and others have seen similar effects after a reduction of MMF and it was shown that a reduction of MMF positively correlated with an immune response in solid organ transplant recipients after vaccination. We have stated this now more carefully in line 352-358 and line 389.

The authors have demonstrated that protracted viral replication/shedding in the presence of anti-spike antibodies was associated with emergence of virions spike proteins that had lesser affinity to the antibody.

Reviewer #4 also attached two papers which are cited below:

Sivasai KS, Mohanakumar T, Phelan D, Martin S, Anstey ME, Brennan DC. Cytomegalovirus immune globulin intravenous (human) administration modulates immune response to alloantigens in sensitized renal transplant candidates. *Clin Exp Immunol*. 2000 Mar;119(3):559-65. doi: 10.1046/j.1365-2249.2000.01138.x. PMID: 10691931; PMCID: PMC1905589.

Sahin U, Muik A, Derhovanessian E, Vogler I, Kranz LM, Vormehr M, Baum A, Pascal K, Quandt J, Maurus D, Brachtendorf S, Lörks V, Sikorski J, Hilker R, Becker D, Eller AK, Grützner J, Boesler C, Rosenbaum C, Kühnle MC, Luxemburger U, Kemmer-Brück A, Langer D, Bexon M, Bolte S, Karikó K, Palanche T, Fischer B, Schultz A, Shi PY, Fontes-Garfias C, Perez JL, Swanson KA, Loschko J, Scully IL, Cutler M, Kalina W, Kyratsous CA, Cooper D, Dormitzer PR, Jansen KU, Türeci Ö. COVID-19 vaccine BNT162b1 elicits human antibody and TH1 T cell responses. *Nature*. 2020 Oct;586(7830):594-599. doi: 10.1038/s41586-020-2814-7. Epub 2020 Sep 30. Erratum in: *Nature*. 2021 Feb;590(7844):E17. PMID: 32998157.

REVIEWERS' COMMENTS

Reviewer #4 (Remarks to the Author):

You have addressed my comments and concerns for alternative/broader mechanisms for viral control associated with modulation of transplant immunosuppression and exogenously provided immunoglobulins. Persistent viral shedding in the face of immune interface/response promotes viral adaptation/evasion. Wish we were smarter.

Thank you.